# Comparative Proteomic Profiling of Ectosomes Derived from Thyroid Carcinoma and Normal Thyroid Cells Uncovers Multiple Proteins with Functional Implications in Cancer

**DOI:** 10.3390/cells11071184

**Published:** 2022-03-31

**Authors:** Magdalena Surman, Sylwia Kędracka-Krok, Magdalena Wilczak, Piotr Rybczyński, Urszula Jankowska, Małgorzata Przybyło

**Affiliations:** 1Department of Glycoconjugate Biochemistry, Faculty of Biology, Institute of Zoology and Biomedical Research, Jagiellonian University in Krakow, 30-387 Krakow, Poland; magdalena.surman@uj.edu.pl (M.S.); magdalena.wilczak@doctoral.uj.edu.pl (M.W.); 2Department of Physical Biochemistry, Faculty of Biochemistry, Biophysics and Biotechnology, Jagiellonian University in Krakow, 30-387 Krakow, Poland; sylwia.kedracka-krok@uj.edu.pl (S.K.-K.); piotr.rybczynski@student.uj.edu.pl (P.R.); 3Proteomics and Mass Spectrometry Core Facility, Malopolska Centre of Biotechnology, Jagiellonian University in Krakow, 30-387 Krakow, Poland; urszula.jankowska@uj.edu.pl

**Keywords:** cancer, ectosomes, LC–MS/MS, proteomics, thyroid carcinoma

## Abstract

Proteins carried by tumor-derived ectosomes play an important role in cancer progression, and are considered promising diagnostic markers. In the present study, a shotgun nanoLC–MS/MS proteomic approach was applied to profile and compare the protein content of ectosomes released in vitro by normal human thyroid follicular epithelial Nthy-ori 3-1 cells and human anaplastic thyroid carcinoma (TC) 8305C cells. Additionally, the pro-migratory and pro-proliferative effects of Nthy-ori 3-1- and 8305C-derived ectosomes exerted on the recipient cells were assessed in wound closure and Alamar Blue assays. A total of 919 proteins were identified in all replicates of 8305C-derived ectosomes, while Nthy-ori 3-1-derived ectosomes contained a significantly lower number of 420 identified proteins. Qualitative analysis revealed 568 proteins present uniquely in 8305C-derived ectosomes, suggesting their applicability in TC diagnosis and management. In addition, 8305C-derived ectosomes were able to increase the proliferation and motility rates of the recipient cells, likely due to the ectosomal transfer of the identified cancer-promoting molecules. Our description of ectosome protein content and its related functions provides the first insight into the role of ectosomes in TC development and progression. The results also indicate the applicability of some of these ectosomal proteins for further investigation regarding their potential as circulating TC biomarkers.

## 1. Introduction

The intercellular transfer of bioactive molecules (mainly proteins, lipids, nucleic acids) by extracellular vesicles (EVs) is one of the crucial mechanisms of cell–cell communication under physiological conditions, as well as in different medical conditions. Over the last few decades, the role of EVs in cancer development and progression have been extensively investigated [1,2,3,4]. In addition to the functional involvement of EVs in carcinogenesis-related processes, a lot of attention has been paid to their molecular contents as a potential source of novel diagnostic targets. After the isolation of EVs from various body fluids (blood, urine, cerebrospinal fluid, etc.,) or conditioned cell culture media, proteomic, transcriptomic, and/or lipidomic profiling was performed in many studies to identify potential cancer biomarkers [5,6,7].

In the recent World Health Organization GLOBOCAN 2020 report, thyroid cancer was ranked 11th among other cancers in terms of morbidity (586,202 cases worldwide), and was responsible for 43,646 deaths in 2020 [8]. The monitoring of traditional markers, such as thyroglobulin or calcitonin, followed by ultrasonography and fine-needle biopsy, is the gold standard for the differential diagnosis of TC. However, this procedure has limitations regarding the distinction between benign and malignant thyroid nodules [9]. Results from EV-oriented research on TC suggest that molecular analysis of EVs from body fluids could be integrated to improve the aforementioned routine clinical approach. In TC, the release of different populations of EVs, i.e., exosomes (having endosomal origin) and ectosomes (membrane-derived, also called microvesicles) was observed, with their amounts significantly increasing in the plasma of TC patients versus the controls [9]. Transcriptomic and proteomic studies have also identified many differentially expressed RNAs and proteins in EVs that play a significant role in TC progression, as well as potential diagnostic markers that could facilitate quicker and more accurate diagnosis in the near future [10].

In contrast to multiple transcriptomic studies related to TC-derived EVs, the amount of research on their protein composition is very limited, and is focused solely on the exosome population. Proteomic analysis of serum exosomes from TC patients revealed the upregulation of multiple proteins, among which, proteins involved in integrin signaling pathways were overrepresented [11]. Furthermore, higher levels of exosomal thyroglobulin [12] and several heat shock proteins [13] were observed in TC patients with advanced tumors versus early-stage tumors or healthy controls. Other studies have demonstrated the functional effect of TC-derived exosomes exerted on recipient cells. Serum exosomes from TC patients interfered with effector T cell activation and suppressed the secretion of inflammatory cytokines [14]. Finally, exosomes derived from in vitro TC cell cultures induced the expression and secretion of selected metalloproteinases by normal fibroblasts [15].

Until now, similar studies have not been conducted on the latter population of EVs, i.e., ectosomes. Nevertheless, tumor-derived ectosomes can also contain a specific protein cargo enriched in cancer biomarkers [16,17,18,19,20]. In the present study, a shotgun proteomic approach was applied to profile and compare the protein content of ectosomes released in vitro by normal human thyroid follicular epithelial Nthy-ori 3-1 cells and human anaplastic thyroid carcinoma 8305C cells. Furthermore, the pro-migratory and pro-proliferative effects of Nthy-ori 3-1- and 8305C-derived ectosomes exerted on recipient cells were examined in wound closure and Alamar Blue assays, respectively.

## 2. Materials and Methods

### 2.1. Materials

RPMI 1640 GlutaMAX™-I medium, fetal bovine serum (FBS), Alamar Blue cell viability reagent, and MicroBCA protein assay kit were obtained from Life Technologies (Carlsbad, CA, USA). Lumi-LightPLUS Western blotting kit (including anti-mouse IgG-HRP secondary antibody), trypsin–EDTA solution, and penicillin–streptomycin solution were purchased from Merck Group (Darmstadt, Germany). The 10% TGX SDS–PAGE stain-free precast gels and 2× concentrated Laemmli buffer were the products of Bio-Rad Laboratories (Hercules, CA, USA). LC–MS grade solvents were supplied by Supelco (Bellefonte, PA, USA). Other chemicals were of analytical grade and commercially available.

### 2.2. Cell Lines and Cell Culture Conditions

The human normal thyroid follicular epithelial cell line Nthy-ori 3-1 and human thyroid anaplastic carcinoma cell line 8305C were kindly provided by Prof. Barbara Czarnocka of the Medical Centre of Postgraduate Education, Warszawa, Poland, and by Prof. Anna Krześlak of the University of Lodz, Poland, respectively. The use of the Nthy-ori 3-1 cell line was reported to the Ministry of the Environment, since they are qualified as a genetically modified microorganism (GMM). Nthy-ori 3-1 cells were immortalized by transfection with 1 μg of the plasmid SV40ori (large T antigen), comprising a 5.3 kB SV40 genome with a six base pair deletion that eliminated the BglI site at the origin of replication cloned in pMK16. T antigen interacts with two protein suppressors of cell division (pRB1 and p53) and inhibits their activity, thus enabling an unlimited number of divisions of transfected Nthy-ori 3-1 cells [21]. Information on genetic alterations of the cell lines are provided in [22,23].

Both Nthy-ori 3-1 and 8305C cells were cultured in RPMI 1640 medium with GlutaMAX-I, supplemented with 10% FBS and antibiotics, i.e., penicillin (100 unit/mL) and streptomycin (100 μg/mL), on 100 mm Petri dishes. The cells were incubated at 37 °C under a humidified atmosphere of 5% CO_2_, and passaged after reaching approximately 80% confluence.

### 2.3. Isolation of Ectosomes

For the preparation of one ectosome sample, cell culturing was carried out until we obtained twenty 100 mm Petri dishes of cells at 80% confluence. Following this, subconfluent Nthy-ori 3-1 and 8305C cells were cultured in serum-free media for 24 h. Next, conditioned media (approx. 200 mL) were collected and subjected to sequential centrifugation. Cells and cellular debris pelleted after centrifugations at 400× *g* (5 min, 4 °C) and 4500× *g* (20 min, 4 °C) were discarded, whereas supernatants were collected and concentrated by a low-vacuum filtration (LVF) procedure, as described in [24]. LVF was performed on the dialysis membrane (Repligen, Waltham, MA, USA, cat. 131486) with MWCO (molecular weight cut-off) of 1000 kDa under a low vacuum (−0.4 bar). The obtained concentrated media (approx. 2 mL) were then centrifuged at 7000× *g* (20 min, 4 °C) to remove larger vesicles, and finally at 18,000× *g* (20 min, 4 °C) to obtain ectosome pellets.

### 2.4. Characterization of Ectosome Samples

Transmission electron microscopy (TEM) [25] and nanoparticle tracking analysis (NTA) [23] were used to verify the purity of the obtained samples. Additionally, both whole-cell protein extracts, prepared as described in [25], and ectosome samples (50 μg of proteins each according to MicroBCA method) were separated on 10% SDS–PAGE stain-free precast gels under reducing conditions, and analyzed by Western blotting using the following primary antibodies: anti-CD63 (1:2000, clone RFAC4, Merck, cat. no. CBL553), anti-HSP70 (1:2000, clone C92F3A-5, Santa Cruz Biotechnology, Dallas, TX, USA, cat. no. sc-66048), and anti-ARF6 (1:500, clone 3A-1, Santa Cruz Biotechnology, cat. no. sc-7971); anti-mouse IgG-HRP (1:400) was used as a secondary antibody. EV markers were detected using chemiluminescent substrates for HRP and ChemiDoc imaging system (Bio-Rad Laboratories).

### 2.5. LC–MS/MS Proteomic Analysis

#### 2.5.1. Sample Preparation for Mass Spectrometric Analysis

The ectosome pellets were washed three times with PBS and suspended in 50 µL of lysis buffer (100 mM Tris-HCl, pH 7.6, 1% SDS). The samples were sonicated for 15 min at 320 W (high intensity setting) with a time interval of 30 s/30 s ON/OFF, using a Bioruptor UCD-200 sonicator (Diagenode, Denville, NJ, USA). Subsequently, samples were incubated at 95 °C for 5 min, and centrifuged (20,000× *g*, 10 min, RT). Supernatants were saved. Protein digestion was performed using S-Trap™ micro spin columns according to the manufacturer’s protocol. Briefly, samples were solubilized in 5% SDS, 50 mM triethylammonium bicarbonate (TEAB, pH 7.55), reduced with 50 mM DTT solution, alkylated with the addition of iodoacetamide to a final concentration of 40 mM, and finally acidified with phosphoric acid. Afterwards, six volumes of S-Trap binding buffer (90% aqueous methanol, 100 mM TEAB, pH 7.1) was added, and the mixture was placed onto S-Trap by centrifugation at 4000× *g* for 10 s. Samples were purified by washing with S-Trap binding buffer and centrifugation. Proteins were digested overnight with trypsin (Promega, Madison, WI, USA) at a 20:1 (wt:wt) protein-to-enzyme ratio. Peptides were eluted with 50 mM TEAB, followed by 0.2% aqueous formic acid and 50% acetonitrile containing 0.2% formic acid. Ultimately, peptides were vacuum-dried. Desalting of peptides was performed using solid-phase extraction on the C18 tips.

#### 2.5.2. Liquid Chromatography and Tandem Mass Spectrometry (LC–MS/MS)

Peptides were analyzed using the UltiMate 3000 RSLCnano system coupled with the Q-Exactive mass spectrometer (Thermo Fisher Scientific, Waltham, MA, USA), as described previously [26]. The samples were resolved on an analytical column (Acclaim PepMap RSLC C18, 75 µm × 500 mm, 2 µm particle, 100 Å pore size) with a 90 min gradient from 2% to 40% acetonitrile in 0.05% formic acid at a flow rate of 200 nL/min. The Q-Exactive was operated in a data-dependent mode using the top eight method. Three technical replicates for each of the two biological samples were measured.

#### 2.5.3. Analysis of Proteomic Data

The RAW files were processed by the Proteome Discoverer platform (v.1.4, Thermo Fisher Scientific) and searched against the SwissProt database with *Homo sapiens* taxonomy restriction (release February 2021, 20,396 sequences) using Mascot search engine (v. 2.5.1, Matrix Science Ltd, London, UK). The following parameters were applied: cysteine carbamidomethylation as fixed modification; methionine oxidation and protein N-terminal acetylation as variable modifications; the peptide mass tolerance was set to 10 ppm and fragment mass tolerance to 20 mmu. Only tryptic peptides with up to one missed cleavage were considered. The false discovery rate for peptide-spectrum matches (PSMs) was set to 0.01 using Percolator [27]. The mass spectrometry data were deposited onto the ProteomeXchange Consortium [28] via the MassIVE repository with the dataset identifier PXD031723.

### 2.6. Bioinformatic Analysis

The final protein lists were created manually, and only include proteins that were identified by both biological repetitions of ectosome samples (that is, in every technical repetition) and with at least two peptides. FunRich 2.0 software with UniProt database (release April 2021) as a reference, was used to create Venn diagrams, including Vesiclepedia protein overlap and gene ontology (GO) analysis, with regard to the cellular compartment, molecular function, and biological processes. For each GO term, 10 categories with the highest statistical significance of protein enrichment within the respective category (calculated as −log10(*p*-value)) are presented as graphs. Interactomes were prepared using string v11.0 (https://string-db.org, accessed on 12 December 2021). Distributed normalized spectrum abundance factors (dNSAFs) were calculated as in [26].

### 2.7. Wound Closure Assay

Nthy-ori 3-1 and 8305C cells were cultured to confluence on six-well plates, and then the cell-coated surface was scraped with a 200 µL pipette tip. Next, two different doses of ectosomes (30 µg and 60 µg of proteins) were added to each well for 18 h. Each wound was photographed in 10 separate fields immediately after scraping (0 h) and after 18 h of incubation. The average rate of wound closure was evaluated by multiple measurements of the wound width using Zeiss, Jena, Germany, AxioVision Rel. 4.8 image analysis software, and calculated as follows:wound closure = (initial wound width (0 h) − wound width after 18 h)/initial wound width (0 h)

Results were expressed as percentages.

### 2.8. Alamar Blue Cell Viability Assay

Nthy-ori 3-1 and 8305C cells were seeded onto 96-well plates at a density of 1 × 10^4^ cells/100 µL. The next day, serum-free medium was added and cells were incubated with two doses of ectosomes (30 µg and 60 µg of proteins) for 18 h. After incubation, 10% of Alamar Blue reagent was added to each well for 2 h, and then the fluorescence intensity was measured at 560/595 nm. Results were standardized in relation to the untreated control (taken as 1).

### 2.9. Statistical Analysis

Alamar Blue and wound closure assays were performed in triplicate for each experimental condition. Analysis of variance (one-way ANOVA) and post hoc Tukey’s tests were later performed with the use of Statistica 12.5 software to test for statistically significant differences, defined by a *p*-value < 0.05.

## 3. Results

### 3.1. Purity of Ectosome Samples

The purity of Nthy-ori 3-1 and 8305C-derived ectosome samples concentrated by LVF followed by 18,000× *g* centrifugation, was evaluated with the use of transmission electron microscopy (TEM) and nanoparticle tracking analysis (NTA). As revealed by TEM images (Figure 1A), neither intact cells, cellular organelles, or other cellular debris were observed in the obtained samples. Isolated populations of ectosomes were moderately heterogeneous in size, and the most numerous subpopulation had a diameter in the range of 100–200 nm and 100–300 nm for Nthy-ori 3-1- and 8305C-derived ectosomes, respectively. Further analysis of the ectosome samples by NTA (Figure 1B) confirmed that the most abundant subpopulations of EVs were those with a diameter in the range of 100–300 nm. Western blotting (WB) analysis of the expression of selected EV protein markers revealed the absence of exosomal markers (i.e., CD63 and HSP70) and simultaneous enrichment in ARF6 (the protein involved in ectosome biogenesis and, for this reason, considered to be an ectosomal marker) in both the obtained EV samples (Figure 1C).

In summary, due to the fact that most of the isolated extracellular microvesicles were in the predefined diameter range for ectosomes (100–1000 nm), and displayed a marker expression pattern that is specific for ectosomes, we considered that contamination of the isolated samples with exosomes was negligible, and the obtained samples were highly enriched in ectosomes.

### 3.2. Proteins Identified in Nthy-ori 3-1- and 8305C-Derived Ectosomes

The comprehensive proteomic profiling of Nthy-ori 3-1- and 8305C-derived ectosomes was performed with the use of gel-free shotgun nanoLC–MS/MS. A total of 919 proteins were identified by at least two peptides in all biological and technical replicates of 8305C-derived ectosomes, while Nthy-ori 3-1-derived ectosomes contained a significantly lower number of 420 identified proteins (Figure 2A). Alongside unique proteins, a shared set of 351 proteins was present in ectosomes released by both cell lines. Complete lists of proteins identified in all particular replicates can be found in Appendix A. The vast majority of proteins identified by the present study (93.6% for both Nthy-ori 3-1- and 8305C-derived ectosomes) were also found by other vesicle-related studies included in the Vesiclepedia database (Figure 2B). This strongly supports their vesicular origin, and proves that they are most likely not a part of co-isolated cell debris or the remnants of conditioned cell culture media.

Furthermore, the lists of identified proteins were subjected to gene ontology (GO) analysis with respect to its three main aspects, i.e., cellular compartment, molecular function, and biological process, using the UniProt database (release April 2021) as a reference. For each analysis, the 10 categories with the highest enrichment are presented as graphs, while the full reports are provided in Appendix A. Regarding the cellular compartment, cytosolic and membrane proteins were the most widely represented groups in both ectosomal samples (up to 63.46% and 42.82%, respectively) (Figure 3). These results correspond with the mechanism of ectosome biogenesis in which fragments of cytoplasm are surrounded by adjacent regions of the plasma membrane. Additionally, for Nthy-ori 3-1-derived ectosomes, a significant percentage of proteins were assigned to the categories “extracellular region” (33.73%) and “extracellular space” (28.47%), reflecting the localization of ectosomes after their release from the cell.

Regarding biological processes, both Nthy-ori 3-1- and 8305C-derived ectosomes were enriched in proteins involved in translation (including the initiation of the process) and different co-translational processes, such as protein targeting or folding (Figure 4). Additionally, both ectosomal samples were rich in proteins linked to neutrophil degranulation. Numerous other GO terms for Nthy-ori 3-1-derived ectosomes were related to extracellular matrix organization, protein metabolism, and glycolysis, as well as platelet degranulation. On the other hand, tumor-derived 8305C-derived ectosomes showed enrichment within the category related to the cancer-promoting Wnt signaling pathway, which reflects their cancerous origin. Additionally, proteins involved in antigen presentation via MHC class I molecules were widely represented in 8305C-derived ectosomes, which may be one of the mechanisms of evading immune system surveillance. Finally, 8305C-derived ectosomes contained proteins related to mRNA splicing, stabilization, and metabolism, suggesting their involvement in post-transcriptional processing.

Classification by molecular function revealed the enrichment in proteins with nucleic acid (e.g., RNA, mRNA, dsRNA) and protein (e.g., cadherin/integrin/unfolded) binding activity in ectosomes from both cell lines (Figure 5). Additionally, 8305C-derived ectosomes contained a significant number of proteins involved in GTP/ATP binding, and proteins with GTPase activity. Therefore, ectosomes may function as indirect coordinators of different signaling pathways in the recipient cells, which often involve nucleotides as signaling molecules.

Next, normalized spectral counting-based quantitative analysis was performed on proteins identified in Nthy-ori 3-1- and 8305C-derived ectosomes, using the methodology described in [29]. Spectral counts of each protein were used to estimate protein abundance in a distributive manner, i.e., peptide spectral counts were calculated for each protein based on unique peptides and the weighted distribution of any peptide shared with homologous proteins. Normalization was performed for protein length and whole-protein content in the sample.

Figure 6 depicts the relative abundance of proteins uniquely identified in 8305C-derived ectosomes (i.e., not identified in any replicate of Nthy-ori 3-1-derived ectosomes). In total, 150 proteins were assigned by GO to exosomal localization, including growth differentiation factor 15 (GDF15), peroxiredoxin-5 (PRDX5), proteasome activator subunit 1 (PSME1), arginosuccinate synthase (ASS), voltage-dependent anion-selective channel protein 1 (VDAC-1), myosin regulatory light chain 12A (MYL12A), adenylate kinase isoenzyme 1 (AK1), proteasome activator complex subunit 2A (PSME2), and carbonyl reductase [NADPH] 1 (CBR1). Although, in the present study, ectosomes (not exosomes) were isolated, a variety of proteins can be found in more than one EV population; thus, the assignment of the aforementioned proteins to exosomal localization by GO only demonstrates their vesicular origin. In addition, 45 of the unique 8305C ectosomal proteins were connected, according to GO, to immune response processes, including interferon-induced GTP-binding protein (Mx1), interferon-induced transmembrane protein 3 (IFITM3), deoxynucleoside triphosphate triphosphohydrolase 1 (SAMHD1), and interferon-induced protein with tetratricopeptide repeats 1 (IFIT1). Within the 10 most abundant proteins, two more proteins were revealed, i.e., ubiquitin cross-reactive protein (ISG15) and signal transducer and activator of transcription 1-alpha/beta (STAT1), the latter of which is the mediator of cellular responses to interferons, other cytokines, and growth factors. Complete results of normalized spectral counting-based abundance factor (dNSAF) analysis are provided in Appendix A.

Finally, diagrams of functional protein association networks were prepared with the use of STRING v. 11.0 software. For Nthy-ori 3-1-derived ectosomes, 5% of the most abundant proteins, based on dNSAFs, that were identified in five or six replicates, were taken into consideration (i.e., 57 proteins). As shown in Figure 7, metabolism-related pathways (i.e., glycolysis/gluconeogenesis, amino acid and carbon metabolism) were the most strongly represented. Analogously, 5% of most abundant proteins from 8305C-derived ectosomes were analyzed, revealing enrichment in proteins related to antigen processing and presentation, as well as cellular response to stress, which corresponds to their cancerous origin (Figure 8). Surprisingly, in both cases, the “infectious disease” category was enriched. There are, however, reports suggesting the involvement of different viruses, such as hepatitis C (HCV) [30], erythrovirus B19 (EVB19) [31], BK virus (BKV) [32], Epstein–Barr virus (EBV) [32,33,34], and human papillomavirus (HPV) [32], in TC pathogenesis. This may explain the presence of these types of proteins in the ectosomes isolated in the present study.

In addition, when all unique proteins from 8305C-derived ectosomes were analyzed by STRING v. 11.0 software, the results showed a significant enrichment in GO categories (Reactome pathways), such as “cytokine signaling in the immune system” (51 proteins, false discovery rate (FDR) = 2.43 × 10^−8^), “interferon signaling” (27 proteins, FDR = 2.43 × 10^−8^), and “interferon alpha/beta signaling” (17 proteins, FDR = 5.41 × 10^−8^) (detailed results in Appendix A). Furthermore, when only 20% of the most abundant unique 8305C ectosomal proteins (i.e., 94 proteins) were analogously analyzed, “type I interferon signaling pathway” was the second most significantly enriched category (FDR = 2.10 × 10^−5^), directly behind the generic term “cellular process”. Eight proteins assigned to this category included Mx1, SAMHD1, STAT1, IFIT1, IFIT3, IFITM3, ISG15, and 2′-5′-oligoadenylate synthetase 3 (OAS3). The potential implications of the ectosomal transfer of these proteins in TC will be further evaluated in the Discussion section.

### 3.3. Functional Effect of Nthy-ori 3-1- and 8305C-Derived Ectosomes on Recipient Cells

TC-derived 8305C ectosomes contained multiple cancer-related proteins, which was revealed by bioinformatics analysis. In Table 1, selected cancer-related GO categories were listed, including proteins involved in cancer cell proliferation, migration, and angiogenesis, as well as in immune and drug responses. Furthermore, functional tests were performed to assess whether the ectosome protein cargo modulates particular processes in recipient cells in vitro. Cell viability and motility were evaluated after 18 h of incubation of Nthy-ori 3-1 and 8305C cells with 8305C-derived ectosomes, as well as with Nthy-ori 3-1-derived ectosomes.

An Alamar Blue cell assay was performed to assess ectosome-induced changes in the viability of recipient cells (Figure 9). The addition of 8305C-derived ectosomes (60 µg proteins) caused an almost 3-fold increase in the measured fluorescence intensity of recipient 8305C cells compared to untreated controls, while the effect of a lower dose (30 µg proteins) was significantly weaker (2.5-fold). A similar dose-dependent effect was observed when 8305C ectosomes were added to Nthy-ori 3-1 cells, but it was significantly weaker (1.5-fold and 1.8-fold for 30 µg and 60 µg doses, respectively). On the contrary, incubation with Nthy-ori 3-1-derived ectosomes had no effect on the viability of either Nthy-ori 3-1 or 8305C cells.

As for the migration abilities of the studied cells (Figure 10), a similar, minimal degree of wound closure (approx. 4–5%) was observed in both cell lines after 18 h of incubation under control conditions (cells not treated with ectosomes) The addition of two doses of tumor-derived 8305C ectosomes, i.e., 30 µg and 60 µg proteins, caused a dose-dependent increase in the wound closure rate for both treated cells; however, Nthy-ori 3-1 cells displayed a weaker response. In the case of ectosomes derived from normal Nthy-ori 3-1 cells, the extent of wound closure by Nthy-ori 3-1 cells only increased when the higher dose was added. The 8305C cells responded more strongly to treatment with Nthy-ori 3-1-derived ectosomes than the Nthy-ori 3-1 cells, suggesting that cancer cells may also be able to utilize normal (physiological) signaling mechanisms, such as EVs, to become self-sufficient, and are more sensitive to these signals than normal cells. However, it is also possible that the observed effect of 8305C-derived ectosomes on the mobility rate of recipient cells might have been a consequence of their increased proliferation, and is not necessarily indicative of their increased migration. However, it cannot be ruled out that this may be the result of the joint action of the two factors mentioned. 

Summing up, ectosomes derived from thyroid carcinoma 8305C cells displayed a higher ability to increase viability and wound closure rate in both recipient cell lines compared to ectosomes derived from normal epithelial Nthy-ori 3-1 cells. Thyroid cancer-derived ectosomes should, therefore, be considered as one of the factors promoting the progression of existing tumors.

## 4. Discussion

### 4.1. TC-Derived Ectosomes Are Enriched in Proteins Involved in Cancer Development and Progression

An increasing amount of evidence supports the potential role of EVs in the pathogenesis of various diseases, including cancer. Tumor-derived EVs can carry a wide variety of biomarkers (mirroring the contents of malignant cells) and, due to the delivery of various signaling molecules, they may regulate the status of the recipient cells. In the present study, proteomic profiling of ectosomes released in vitro by 8305C TC and Nthy-ori 3-1 normal thyroid epithelial cells was performed with the use of nanoLC–MS/MS. Such functional characterization of TC-derived ectosomal content provides valuable information regarding the mechanisms of TC development and progression.

As shown in previous studies concerning TC-derived exosomes, this subpopulation of EVs can transfer signaling proteins from malignant to neighboring nonmalignant cells. For instance, Li et al. [35] reported that annexin A1 (ANXA1) is transferred from thyroid squamous cell carcinoma SW579 cells to Nthy-ori 3-1 cells via exosomes. ANXA1 is a 37 kDa calcium- and phospholipid-binding protein that participates in carcinogenesis and tumor progression due to its ability to modulate various cancer-related pathways [36]. ANXA1 transfer resulted in the promotion of proliferation, invasion, and epithelial-to-mesenchymal transition of Nthy-ori 3-1 cells, as well as tumor growth, in a xenograft mouse model. These findings indicate that exosomal ANXA1 can promote TC development and malignant transformation, and might be a target in the development of novel therapies in TC. Our present studies also showed the presence of ANXA1 in 8305C-derived ectosomes (alongside several other members of the annexin protein family); however, the exact role of ectosomal ANXA1 in the cell–cell communication between TC and normal thyroid cells is yet to be assessed.

It has been demonstrated that TC-derived EVs can also regulate the invasion and metastasis of recipient TC cells. For instance, sera-derived exosomes from TC patients with lymph node metastases (compared to non-metastatic TC) increased the invasiveness of BHT101 thyroid cancer cells in vitro in a transwell invasion assay [11]. The effect was attributed to the presence of multiple integrins, including α2, α2b, αν, β1, β2, and β3 subunits, along with the proteins located upstream of the integrin pathway, i.e., talin-1 (TLN1), calpain small subunit 1 (CAPNS1), and proto-oncogene tyrosine protein kinase Src (SRC). In the present study, the β1 integrin subunit and talin-1 were identified in both Nthy-ori 3-1- and 8305C-derived ectosomes. Regarding other integrins, α3 subunit was also found in all samples. In contrast, SRC and CAPNS1 were only present in tumor-derived 8305C ectosomes, suggesting that these ectosomes may play an important role in the regulation of integrin-mediated cell adhesion during tumor dissemination. Moreover, GDF15, one of the mitochondrial cytokines (mitokines), was the second most abundant protein identified uniquely in 8305C-derived ectosomes. GDF15 expression increases upon mitochondrial stress, and was linked to higher viability, invasiveness, and migration rates of different TC cell lines compared to control cultures with a silenced GDF15 gene [37]. The same study showed that GDF15-induced STAT3 activation was an important determinant of TC progression/aggressiveness in patients, suggesting that the GDF15–STAT3 signaling axis may be a novel therapeutic target. The presence of all aforementioned proteins in 8305C-derived ectosomes is also a possible explanation for the cancer-promoting effect observed in the present study in the wound closure and Alamar Blue assays. Tumor-derived 8305C ectosomes enhanced the viability and wound closure rate of recipient cells to a greater extent than ectosomes from normal epithelial Nthy-ori 3-1 cells. Therefore, TC-derived ectosomes, and their protein cargo, should be considered as one of the factors promoting tumor growth, as well as the invasion and migration of tumor cells.

Evidently, EVs are an integral part of the tumor microenvironment, and facilitate the mutual interactions between tumor cells and fibroblasts, vascular cells, immune cells, etc. The particular components of EVs can also affect the extracellular matrix structure, enabling tumor migration/invasion, angiogenesis, and/or leading to the formation of premetastatic niches in distant organs. It has been demonstrated that EVs (an unseparated pool, obtained by ultracentrifugation) isolated from co-cultures of TC cells and fibroblasts, promote invasive phenotypic alterations in recipient cells [15]. After incubation with EVs, increased secretion of inactive zymogen of metalloproteinase 2 (proMMP2) and the active form of MMP2 from recipient TC cells, as well as from normal fibroblasts, was observed. Significant CD147 (MMP activator, known as basigin) expression was also demonstrated in EVs, revealing the importance of TC cell–fibroblast interaction in the formation of specialized EVs, and in the preparation of a microenvironment suitable for TC progression. In the present study, both 8305C- and Nthy-ori 3-1-derived ectosomes contained CD147 and MMP2, but none of the other MMPs. The presence of tissue inhibitors of matrix metalloproteinases 1 and 2 (TIMP-1 and TIMP-2) was also confirmed, further suggesting a complex regulatory role of ectosomes within the tumor microenvironment.

The development of a vascular network within and around growing tumors is the crucial step for subsequent tumor cell invasion and metastasis. EVs are well-known carriers of multiple pro-angiogenic factors that, upon delivery to recipient endothelial cells, stimulate the formation of new blood vessels. Regarding TC-derived EVs, exosomes derived from the hypoxic BCPAP TC cell line were shown to induce an angiogenic phenotype in HUVEC cells [38]. An increase in endothelial tube formation on Matrigel-coated plates was observed, along with increased migration in transwell and wound closure assays. Furthermore, the proliferation rate of exosome-treated HUVEC cells was higher in the CCK-8 assay. Regarding ectosomes, their role in TC-related angiogenesis is yet to be determined. In the present study, no functional tests using endothelial cells were performed; however, multiple pro-angiogenic proteins were identified during GO analysis. In the case of tumor-derived 8305C ectosomes, this category includes 23 proteins, already shown in Table 1. Surprisingly, they did not include any of the classical pro-angiogenic factors, such as vascular endothelial growth factor (VEGF), platelet-derived growth factor (PDGF), angiopoietins, tumor necrosis factor α (TNF-α), or interleukin-6 (IL-6) [39]. This suggests that TC-derived EVs might utilize alternative signaling pathways while regulating tumor angiogenesis.

Cancer progression is also associated with increased platelet activation, aggregation, and degranulation. The activity of thyroid hormones on platelets has already been linked to blood clotting, mainly through L-thyroxine (T4), the principal ligand for αvβ3 integrin on tumor cells and platelets. T4 activates the integrin, promotes platelet aggregation and degranulation (through local ATP release), and stimulates tumor cell proliferation [40]. In 8305C-derived ectosomes, we identified the αv integrin subunit, which may potentially be transferred via ectosomes to facilitate interactions between cancer cells, and between cancer cells and platelets. Moreover, 8305C-derived ectosomes contained multiple other procoagulant (e.g., plasminogen activator inhibitor 1, thrombospondin-1) as well as anti-coagulant (e.g., urokinase-type plasminogen activator, tissue-type plasminogen activator) factors, indicating a more complex role of ectosomes in blood coagulation during cancer progression. Nevertheless, it was the Nthy-ori 3-1-derived ectosomes that showed significant enrichment in the “platelet degranulation” GO category (Figure 4). This suggests that thyroid-derived ectosomes may also regulate blood coagulation under physiological conditions.

Tumor-derived EVs can alter the immune response towards cancer cells either by the activation or suppression of the immune system. Similarly, EVs released by immune cells, such as B cells, regulatory T cells, or tumor-associated macrophages, can suppress cytotoxic T cells and other effector cells. In the case of TC, sera-derived EVs enriched in programmed death protein 1 (PD-1) and programmed death ligand 1 (PD-L1) suppressed the activation of recipient effector CD8+ T cells and decreased the levels of inflammatory cytokines (CD69, interferon γ, and TNF-α) secreted by these cells [14]. In the present study, tumor-derived 8305C ectosomes did not contain PD-L1. Nevertheless, multiple proteins were assigned to the GO category “immune response”, including thrombospondin-1, peroxidasin homolog, complement system components C1r and C3, and HLA class I histocompatibility antigens A (alpha chain) and B (alpha chain). Via the transfer of components from complementary systems, ectosomes may contribute to the induction of a cancer-promoting, pro-inflammatory state within the tumor microenvironment. On the other hand, EV-mediated loss of HLA antigens, essential for recognition by immune cells, may favor the evasion of tumor cells from immune system surveillance.

When investigating the role of tumor-derived ectosomes in immune responses, particular attention should be paid to the proteins involved in cytokine signaling. Cytokines can regulate the proliferation and migration of TC cells, and several are currently under evaluation as potential predictive, diagnostic, and prognostic markers in TC, as well as therapeutic targets [41]. As mentioned in the Results section, unique proteins from 8305C-derived ectosomes showed significant enrichment in such GO categories as “cytokine signaling in the immune system”, “interferon signaling”, and “interferon alpha/beta signaling”. Additionally, when only 20% of the most abundant unique 8305C ectosomal proteins were analyzed, “type I interferon signaling pathway” was the second most significantly enriched category, with eight proteins assigned, i.e., Mx1, SAMHD1, STAT1, IFIT1, IFIT3, IFITM3, ISG15, and OAS3. Most of these proteins have already been implicated in different types of cancer [42,43,44], though there are few connections to TC. STAT1, a transcription factor, was shown to regulate the expression of oncogenic long non-coding RNAs, for instance, LINP1 [45] and ZFPM2-AS1 [46], in TC. As a result, cancer-promoting effects, such as increased cancer cell proliferation, migration, and invasion, and the inhibition of apoptosis, were observed. Furthermore, ISG15 expression was successfully used to distinguish between non-metastatic and lymph node metastatic TC, while in mice, knock down of ISG15 inhibited xenografted tumor growth [47]. In another study, the expression of ISG15 was also correlated with the level of capsular infiltration of immune cells in TC tumors [48].

Finally, GO analysis related to “biological process” (Figure 4) revealed the enrichment in proteins related to neutrophil degranulation in both Nthy-ori 3-1- and 8305C-derived ectosomes. In general, neutrophils contain different molecular factors involved in thyroid hormone metabolism and function, such as thyroid hormone receptors, transporters, and deiodinases. On the other hand, thyroid hormones exert a pro-inflammatory effect in neutrophils, including the production of reactive oxygen species (ROS) and induction of myeloperoxidase (MPO) activity [49]. In human TC samples, neutrophil density correlated with tumor size, while in vitro studies showed that neutrophils are recruited by TC cells through the release of CXCL8/IL [50]. Moreover, the release of granulocyte colony-stimulating factor (GM-CSF) by TC cells improved neutrophil survival. Upregulated proinflammatory activities, and the expression of selected tumor-promoting factors, were also observed in neutrophils. The precise mechanisms driving the mutual interactions between normal/cancerous thyroid cells and neutrophils are yet to be examined. Considering the multiple proteins related to neutrophil degranulation that were identified in the present study, the involvement of EVs, including ectosomes, should be one of the main focal points in further studies.

### 4.2. Clinical Relevance of Proteins Identified in TC-Derived Ectosomes

Once EVs are released into the intracellular space, and further into body fluids, their protein cargo retains the partial characteristics of parental cells. Some proteins directly linked to EV biogenesis, or binding with recipient cells, are commonly found in EVs, regardless of their source. On the other hand, more specific proteins can be identified in EVs that precisely reflect the cellular origin of the EVs, as well as the state of the parental cell, including carcinogenesis or metastasis, etc. With regard to this, and to their wide bioavailability, EVs should be considered as a potential source of novel biomarkers for different diseases, including TC. EVs recovered through non-invasive liquid biopsies (e.g., venous blood and urine sampling) could be used to complement often inaccurate cytological tests, and reduce the need for invasive needle biopsies.

In previous research, some potential protein biomarkers in TC-derived EVs were indicated. For instance, the levels of thyroglobulin in urinary exosomes from TC-patients at metastatic T3 stage were higher than in those at T1 and T2 stages [12]. Furthermore, thyroglobulin levels in urinary exosomes increased after tumor removal in patients with high risk of recurrence, proving exosomal thyroglobulin as a sensitive prognostic and postoperative recurrence marker in TC. However, thyroglobulin was not identified in Nthy-ori 3-1- and 8305C-derived ectosomes in the present study. Its ceased secretion/release was most likely caused by the simplicity of in vitro monoculture of the thyroid cells, and the lack of typical cell–cell interactions in the microenvironment.

In other studies, the expression of chaperones HSP27, HSP60, HSP70, and HSP90 was analyzed by Western blotting performed on plasma exosomes of TC-patients versus those with benign tumors [13]. Alongside HSP70, the expression of three other chaperones was higher in the exosomes of TC patients. Additionally, their expression decreased after tumor-removal surgery, directly indicating their tumor origin and potential clinical application in assessing surgery effectiveness. Furthermore, PD-L1 and PD-1 levels in plasma exosomes from pediatric TC patients were found to be higher in comparison to healthy controls, and PD-L1 expression correlated with T staging [14]. Consistent with the results from the aforementioned studies on chaperones [13], levels of exosomal PD-L1 were found to drop post-surgery. In the present study, PD-1, PD-L1, and HSP27 were not found in tumor-derived 8305C ectosomes. However, 8305C-derived ectosomes were the carriers of many other chaperones from various families, such as HSP10, HSP60, HSP70, HSP71, HSP90, and HSP105. Importantly, the expression of heat shock proteins increases under stress conditions, such as hypoxia or ischemia, both of which occur during tumor growth. It is well known that heat shock proteins are involved in malignant transformation, metastasis, and multidrug-resistance (by providing protection against therapy-induced apoptosis) [51]. Multiple heat shock proteins have already been shown to be differentially expressed in TC, including HSP27, HSP60, HSP0, and HSP90, and to indicate poor prognosis in terms of survival and response to therapy [52]. Therefore, ectosomal chaperones should also be evaluated in terms of their potential clinical applications.

Notably, all the aforementioned studies focused solely on exosome populations, and did not utilize any proteomic techniques. To our knowledge, the only proteomic study concerning TC-derived EVs was performed by Luo et al. [11], analyzing serum exosomes from papillary thyroid cancer patients with and without lymph node metastasis, and from healthy donors, using LC–MS/MS with tandem mass tag label quantification. In the exosomes from patients with lymph node metastasis, 697 differentially expressed proteins were identified, and overexpressed proteins included SRC, TLN1, integrin β2 subunit, and CAPNS1. As already mentioned in Section 4.1, all of these proteins were also present in tumor-derived 8305C ectosomes, suggesting that this population of EVs could also assist the diagnosis and prognosis of TC along with the recurrence and metastasis prediction.

In addition to the diagnostic and prognostic applications of EVs in TC, EVs can be utilized as therapeutic targets. Due to their low immunogenicity and toxicity, and their high biocompatibility and membrane permeability, EVs may serve as delivery vehicles of anti-cancer drugs to precise tumor locations. Gangadaran et al. [53] demonstrated that EVs released by CAL62 TC cells labeled with Renilla luciferase were internalized into a CAL62 tumor in a mouse model 30 min after injection. Moreover, EVs can contribute to the development of novel immunotherapeutic strategies. NK cell-derived EVs after incubation with IL-15 were shown to increase cell lysis and inhibit the growth of some human cancer cell lines, including TC, but did not affect normal cells [54]. Finally, human adipose-derived stem cells were used to deliver synthetic tyrosine kinase inhibitor (TKI) to enhance iodine avidity in radioactive iodine-refractory SW1736 thyroid cancer cells [55]. The expression of mRNAs and proteins involved in iodide-metabolism was higher in the (EVs+TKI)-treated than in the free TKI-treated SW1736 cells. Furthermore, EVs+TKI treatment enhanced 125I uptake by SW1736 cells, suggesting that this approach can reverse radioiodine-resistance in thyroid cancer cells. Tyrosine kinase inhibitors were not identified in ectosomes in the present study; however, this population of EVs can also be loaded with particular therapeutic agents, including proteins and chemotherapeutics.

## 5. Conclusions

Despite the fact that only a few studies have been performed on EVs isolated either from body fluids of TC patients, or from TC in vitro cell cultures, evidence exists that suggests their involvement in TC development and progression. Moreover, TC-derived EVs could be used to develop new, or to improve current, strategies for the diagnosis and treatment of TC. Primarily, an EV-based liquid biopsy should be widely implemented as an alternative and complementary diagnostic method in cases where a cytological diagnosis is uncertain. In the present study, specific alterations to the proteomic profile have been reported, for the first time, in ectosomes derived from 8305C TC cells. Our description of the complex ectosome protein content, and its related functions, provides preliminary insight into the role of this population of EVs in disease biogenesis and pathogenesis. Our results also indicate the applicability of some of these ectosomal proteins for further investigation regarding their potential as circulating TC biomarkers. 

We acknowledge the limitations of our current study. Although our study is the first to analyze the protein content of thyroid-derives ectosomes, two established cell lines that were not derived from the same donor were used. Furthermore, although Nthy-ori 3-1 cells are commonly described as normal human primary thyroid follicular epithelial cells, and are regularly used for the study of control and growth in human thyroid, they do not completely reflect normal thyroid cells, as they were transfected with a plasmid containing an origin-defective SV40 genome (SV-ori) to immortalize them, alongside the retention of several different functions [56]. Thus, evaluating changes in ectosome cargo in relation to thyroid cancer using a wider range of thyroid cancer cell lines; for instance, representing different disease stages or more differentiated cancers; could provide a better understanding of the role of EVs in thyroid cancer progression. Additionally, primary cultures established from patients ’histologically non-tumorous/tumorous tissues could also be used in future studies to provide more a clinically-relevant reference.

## Figures and Tables

**Figure 1 cells-11-01184-f001:**
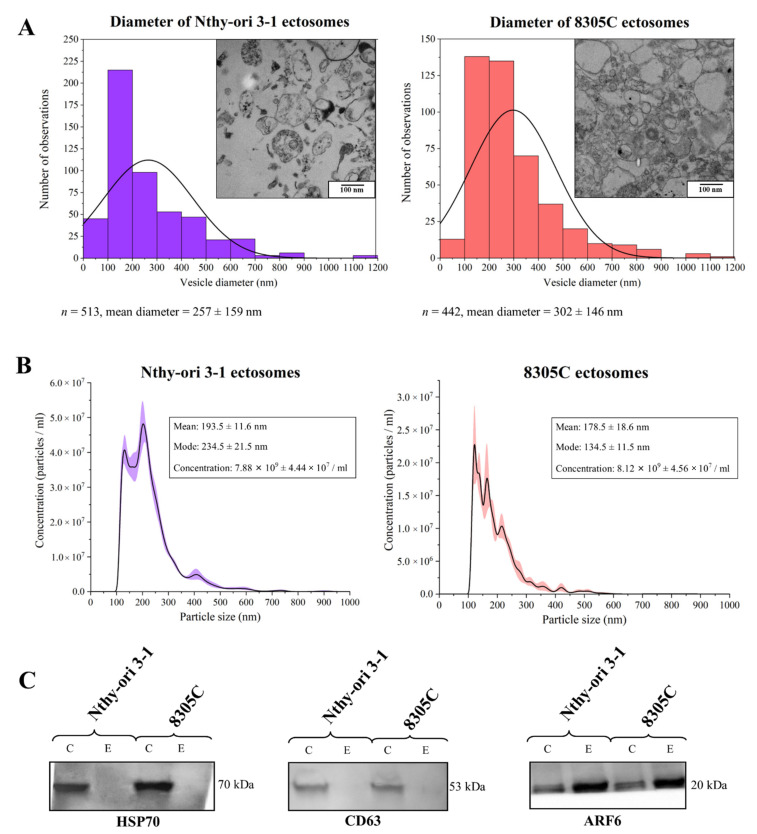
Characterization of ectosome samples isolated from conditioned media of normal thyroid follicular epithelial Nthy-ori 3-1 cells and thyroid anaplastic carcinoma 8305C cells. (**A**) Morphological characterization of Nthy-ori 3-1- and 8305C-derived ectosomes by transmission electron microscopy (TEM). Size distributions are presented on histograms. Mean diameter ± standard deviation was calculated for all observed vesicles (n) from a respective sample. (**B**) Nanoparticle tracking analysis (NTA) of Nthy-ori 3-1- and 8305C-derived ectosomes. Results from five independent measurements for each cell line are presented as graphs. The shaded area depicts standard deviation. (**C**) Representative Western blot of extracellular vesicle markers in whole-cell protein extracts (line C) and ectosome samples (line E). After being separated by 10% SDS–PAGE and transferred into PVDF membrane, 50 μg of proteins were probed with anti-HSP70 (1:2000), anti-CD63 (1:2000), and anti-ARF6 (1:500) as primary antibodies, and anti-mouse IgG-HRP (1:400) as a secondary antibody.

**Figure 2 cells-11-01184-f002:**
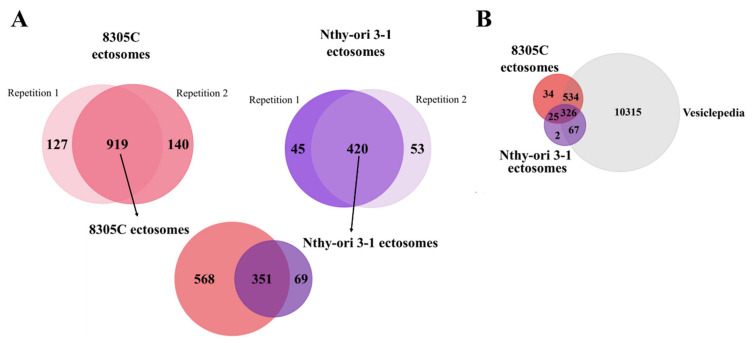
(**A**) Number of proteins identified in particular numbers of replicates of ectosome samples (the total number of six replicates included two biological replicates with three technical replicates each) and Venn diagram illustrating the number of proteins identified in two biological replicates of ectosomes derived from normal thyroid follicular epithelial Nthy-ori 3-1 cells and thyroid anaplastic carcinoma 8305C cells by at least two peptides. (**B**) Venn diagram illustrating protein overlap between isolated ectosomes with Vesiclepedia database as a reference.

**Figure 3 cells-11-01184-f003:**
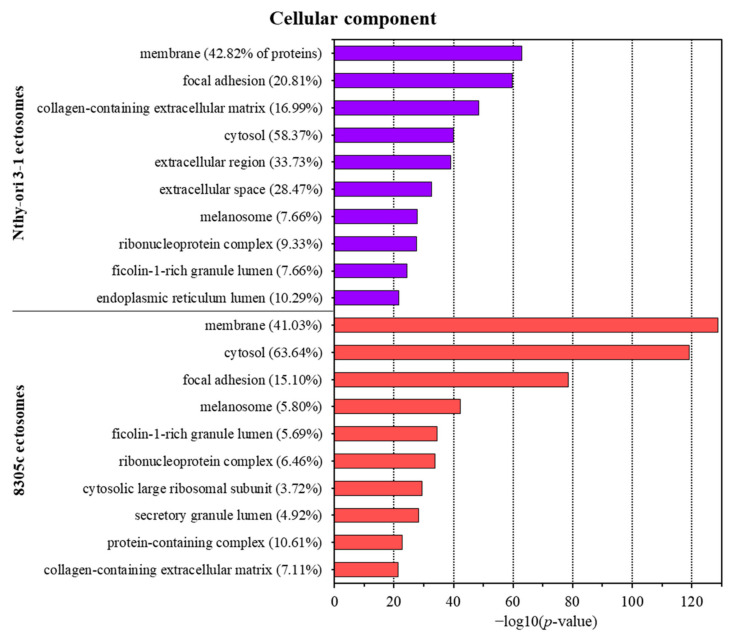
Gene ontology (GO) analysis of proteins identified in ectosomes derived from normal thyroid follicular epithelial Nthy-ori 3-1 cells and thyroid anaplastic carcinoma 8305C cells performed with the use of FunRich 2.0 software with UniProt database (release April 2021) as a reference. For the GO term “cellular compartment”, 10 categories with the highest statistical significance of protein enrichment within the specific category (*p* < 0.001) are presented as graphs. Complete results of GO analysis are provided in Appendix A.

**Figure 4 cells-11-01184-f004:**
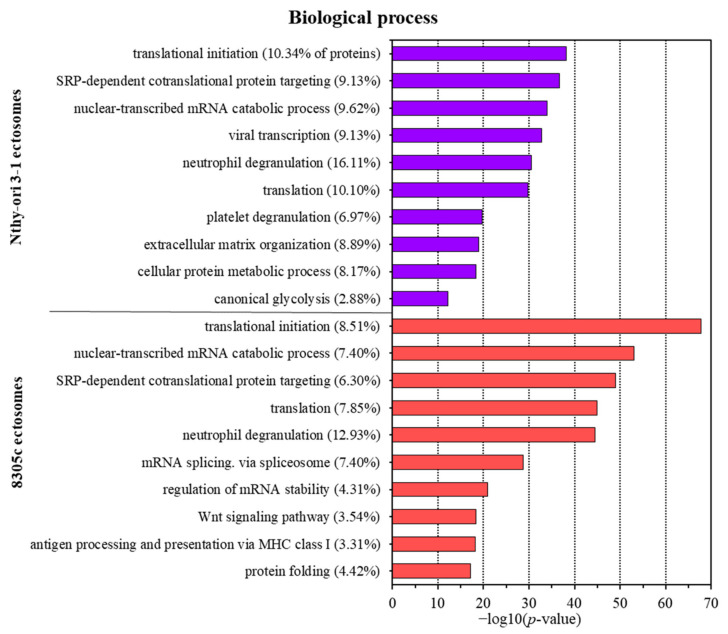
Gene ontology (GO) analysis of proteins identified in ectosomes derived from normal thyroid follicular epithelial Nthy-ori 3-1 cells and thyroid anaplastic carcinoma 8305C cells performed with the use of FunRich 2.0 software with UniProt database (release April 2021) as a reference. For GO term “biological process”, 10 categories with the highest statistical significance of protein enrichment within the specific category (*p* < 0.001) are presented as graphs. Complete results of GO analysis are provided in Appendix A.

**Figure 5 cells-11-01184-f005:**
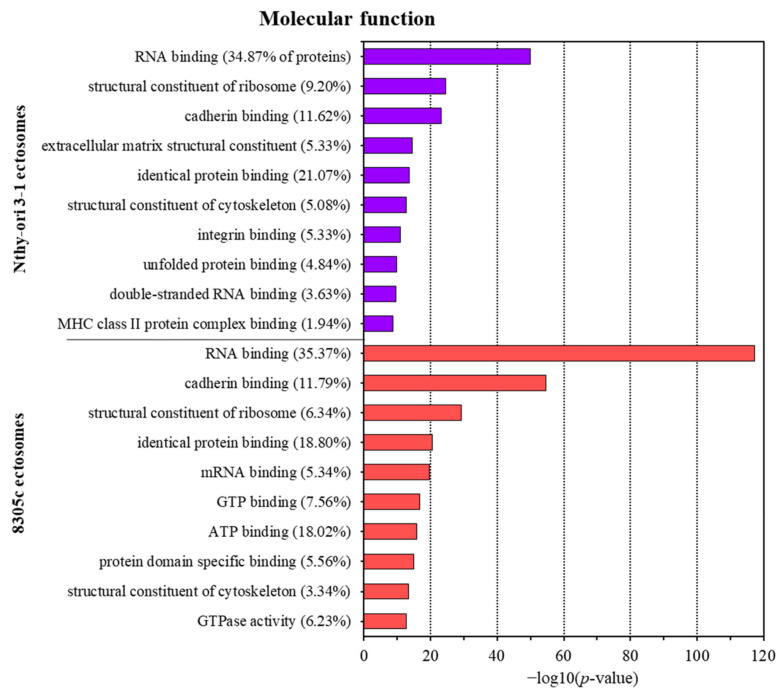
Gene ontology (GO) analysis of proteins identified in ectosomes derived from normal thyroid follicular epithelial Nthy-ori 3-1 cells and thyroid anaplastic carcinoma 8305C cells performed with the use of FunRich 2.0 software with UniProt database (release April 2021) as a reference. For GO term “molecular function”, 10 categories with the highest statistical significance of protein enrichment within the specific category (*p* < 0.001) are presented as graphs. Complete results of GO analysis are provided in Appendix A.

**Figure 6 cells-11-01184-f006:**
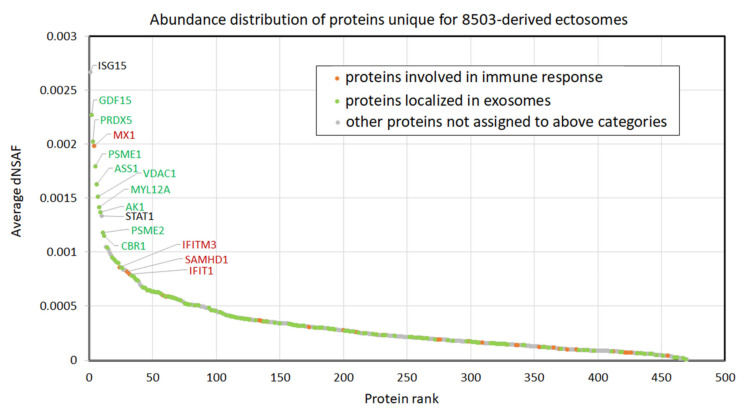
Normalized spectral counting-based abundance of proteins identified uniquely in 8305C-derived ectosomes (not identified in any replicate of Nthy-ori 3-1-derived ectosomes). dNSAF, distributed normalized spectrum abundance factor. Complete results of analysis are provided in Appendix A.

**Figure 7 cells-11-01184-f007:**
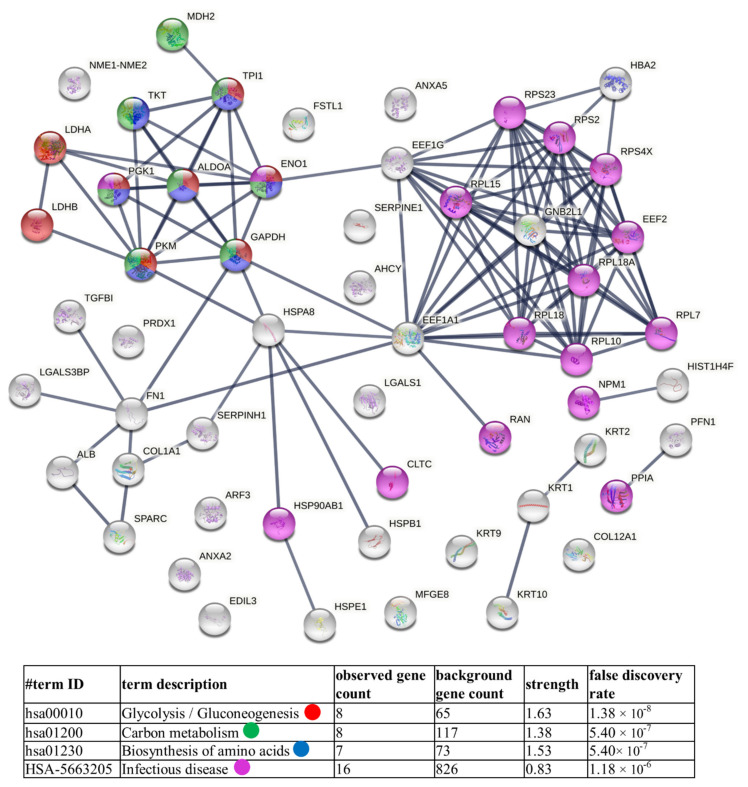
Diagram of functional protein association networks prepared with the use of STRING v. 11.0 software for 5% of the most abundant proteins identified in Nthy-ori 3-1-derived ectosomes (i.e., 57 proteins). The selected strongly represented pathways are presented as an interactome. Complete results are provided in Appendix A.

**Figure 8 cells-11-01184-f008:**
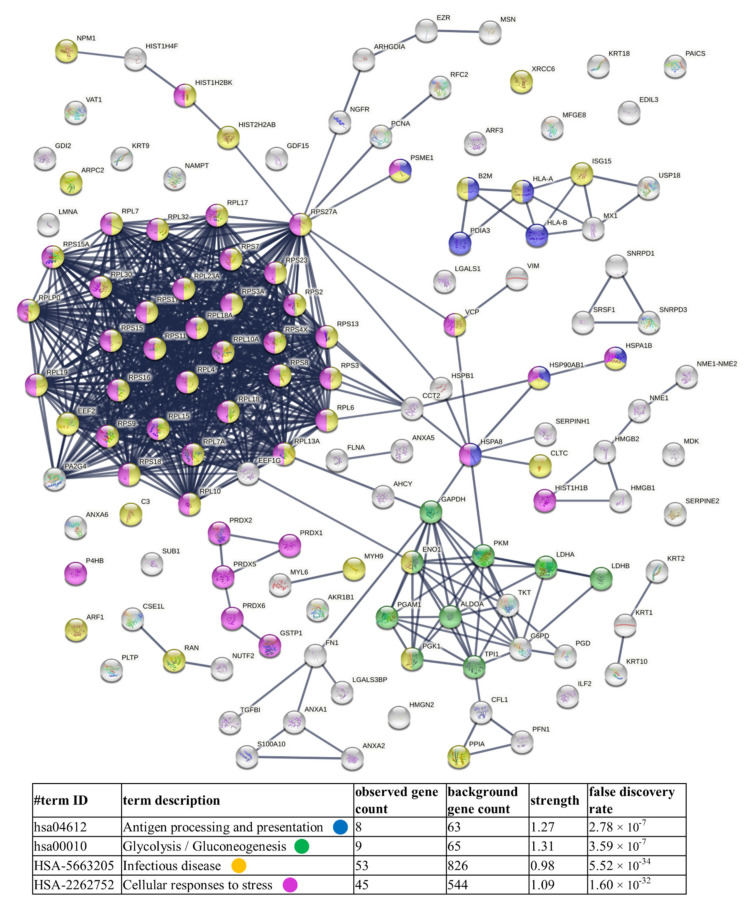
Diagram of functional protein association networks prepared with the use of STRING v. 11.0 software for 5% of the most abundant proteins identified in 8305C-derived ectosomes (i.e., 127 proteins). The selected strongly represented pathways are presented as an interactome. Complete results are provided in Appendix A.

**Figure 9 cells-11-01184-f009:**
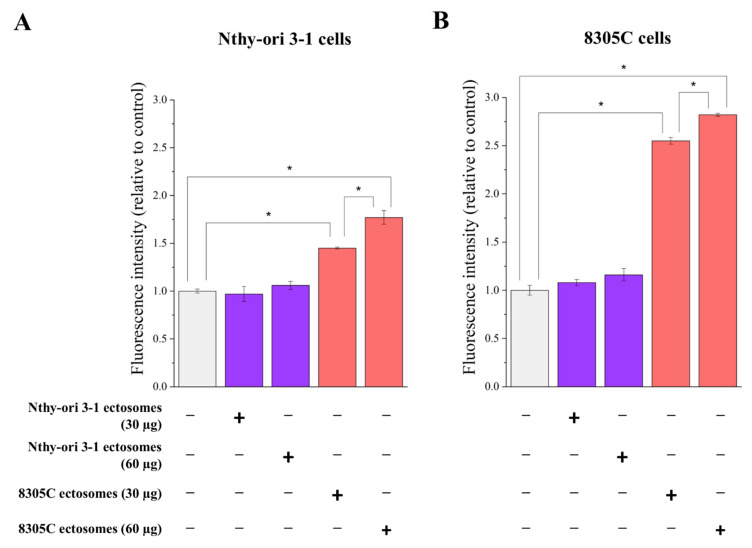
The effect of incubation of normal thyroid follicular epithelial Nthy-ori 3-1 cells (**A**) and thyroid anaplastic carcinoma 8305C cells (**B**) with ectosomes released by these cells. Alamar Blue cell viability assay was carried out after 18 h of incubation with ectosomes. All experiments were conducted in triplicate. “*” denotes statistically significant differences (Tukey’s post hoc test, *p*-value < 0.05).

**Figure 10 cells-11-01184-f010:**
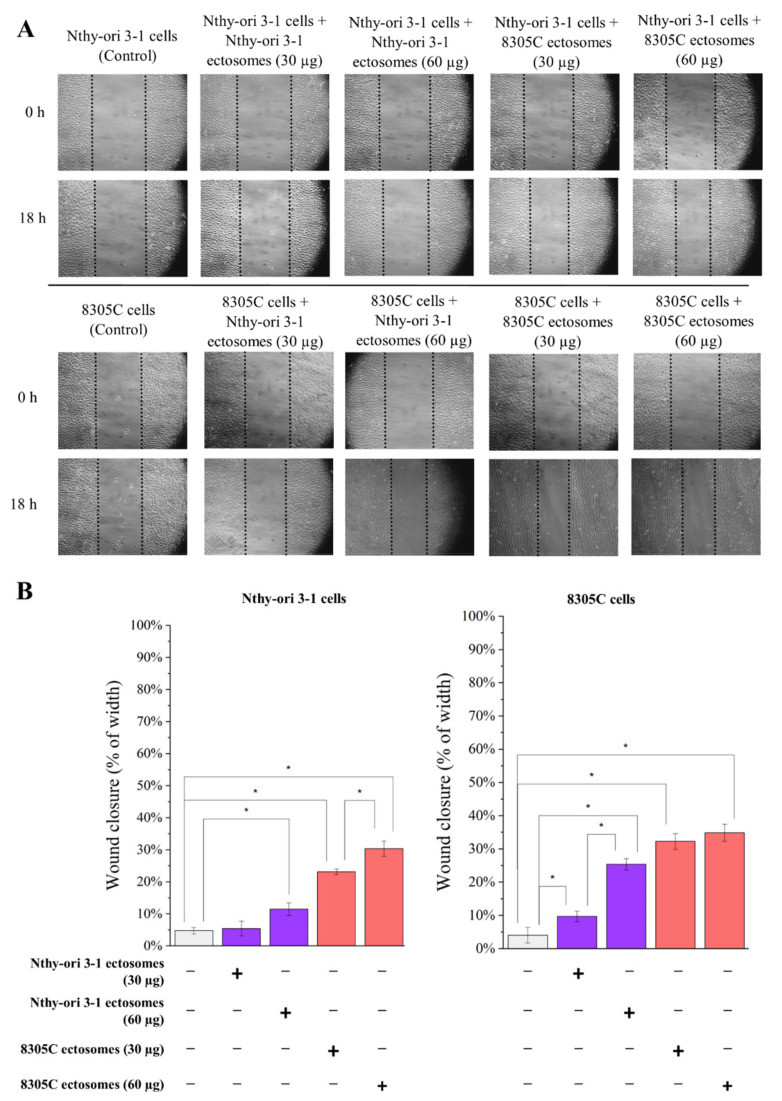
The effect of incubation of normal thyroid follicular epithelial Nthy-ori 3-1 cells and thyroid anaplastic carcinoma 8305C cells with ectosomes released by these cells. Wound closure assay was performed after 18 h of incubation with ectosomes. (**A**) Representative images were taken at 0 h and at 18 h. (**B**) Graphs presenting percentage of wound width closure calculated from three replicates. “*” denotes statistically significant differences (Tukey’s post hoc test, *p*-value < 0.05).

**Table 1 cells-11-01184-t001:** Functional classification of cancer-related proteins identified in ectosomes derived from 8305C human thyroid anaplastic carcinoma cells.

Cell Proliferation (GO:0008283)	Cell Migration (GO:0016477)	Angiogenesis (GO:0001525)	Immune Response (GO:0006955)	Cellular Response to Drug (GO:0035690)
Cyclin-dependent kinase 1 (CDK1) Guanine nucleotide-binding protein G(i) subunit alpha-2 (GNAI2) Guanine nucleotide-binding protein G(I)/G(S)/G(T) subunit beta-1 (GNB1) H3 histone, family 3B (H3F3B ) Lysosomal acid lipase/cholesteryl ester hydrolase (LIPA) Minichromosome maintenance complex component 7 (MCM7) Myosin-10 (MYH10) Peroxiredoxin-1 (PRDX1) Protein scribble homolog (SCRIB) Proto-oncogene tyrosine protein kinase Src (SRC) Ras-related C3 botulinum toxin substrate 1 (RAC1) Ras-related protein Rap-1b (RAP1B) Thioredoxin reductase 1, cytoplasmic (TXNRD1) Transforming growth factor-beta-induced protein ig-h3 (TGFBI) Ubiquitin carboxyl-terminal hydrolase isozyme L1 (UCHL1) Ubiquitin-conjugating enzyme E2L 3 (UBE2L3) X-ray repair cross-complementing protein 5 (XRCC5)	ADP-ribosylation factor 4 (ARF4) Asparagine-tRNA ligase, cytoplasmic (NARS1) CD151 antigen (CD151) CD44 antigen (CD44) CD63 antigen (CD63) Cell division control protein 42 homolog (CDC42) Collagen alpha-1(V) chain (CO5A1) Coronin-1B (CORO1B) Coronin-1C (CORO1C) Cullin-3 (CUL3) Cyclin-dependent kinase 1 (CKD1) Ephrin type-A receptor 2 (EPHA2) Fascin (FSCN1) Glypican-1 (GPC1) Guanine nucleotide exchange factor VAV2 (VAV2) Inactive tyrosine protein kinase 7 (PTK7) Integrin beta-1 (ITB1) Junction plakoglobin (PLAK) Laminin subunit alpha-5 (LAMA5) Laminin subunit beta-1 (LAMB1) Laminin subunit beta-2 (LAMB2) Laminin subunit gamma-1 (LAMC1) Microtubule-associated protein RP/EB family member 1 (MAPRE1) Nck-associated protein 1 (NCKP1) Neural cell adhesion molecule L1 (L1CAM) Protein scribble homolog (SCRIB) Ras GTPase-activating-like protein IQGAP1 (IQGAP1) Ras-related protein Rab-1A (RAB1A) Rho-related GTP-binding protein RhoA (RHOA) Thrombospondin-1 (TSP1)	14-3-3 protein zeta/delta (YWHAZ) 72 kDa type IV collagenase (MMP2) Actin, cytoplasmic 2 (ACTG1) Aminoacyl tRNA synthase complex-interacting multifunctional protein 1 (AIMP1) Annexin A2 (ANXA2) ATP synthase subunit beta, mitochondrial (ATP5F1B) Basement membrane-specific heparan sulfate proteoglycan core protein (HSPG2) Caveolin-1 (CAV1) Chloride intracellular channel protein 4 (CLIC4) Collagen alpha-1(XVIII) chain (COL18A1) Collagen alpha-2(IV) chain (COL4A2) E3 ubiquitin-protein ligase RNF213 (RNF213) Fibronectin (FN1) Guanine nucleotide exchange factor VAV2 (VAV2) Integrin alpha-V (ITGAV) Lactadherin (MFGE8) Myosin-9 (MYH9) Nucleolin (NCL) Peptidyl-prolyl cis/trans isomerase H (PPIH) Plasminogen activator inhibitor 1 (PAI1) Programmed cell death protein 6 (PDCD6) Transforming growth factor-beta-induced protein ig-h3 (TGFBI) Tryptophan-tRNA ligase, cytoplasmic (WARS1)	Complement C1r subcomponent (C1R) Complement C3 (C3) Deoxynucleoside triphosphate triphosphohydrolase (SAMHD1) HLA class I histocompatibility antigen, A alpha chain (HLA-A) HLA class I histocompatibility antigen, B-7 alpha chain (HLA-B) HLA class I histocompatibility antigen, Cw-7 alpha chain (HLA-C) Interferon-induced transmembrane protein 3 (IFITM3) Peroxidasin homolog (PXDN) Purine nucleoside phosphorylase (PNP) Thrombospondin-1 (THBS1)	116 kDa U5 small nuclear ribonucleoprotein component (EFTUD2) 78 kDa glucose-regulated protein (HSPA5) CAD protein (CAD) Cellular tumor antigen p53 (TP53) Growth arrest-specific protein 6 (GAS6) Heat shock protein 90 alpha family class b member 1 (HSP90AB1) Nucleoside diphosphate kinase A (NME1) Ras-related protein Rap-1b (RAP1B) Transferrin receptor protein 1 (TFRC) Ubiquitin carboxyl-terminal hydrolase isozyme L1 (UCHL1)

## Data Availability

MS/MS data are available via ProteomeXchange with identifier PXD031723.

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
