# Peer review of "Comparative Proteomic Profiling of Ectosomes Derived from Thyroid Carcinoma and Normal Thyroid Cells Uncovers Multiple Proteins with Functional Implications in Cancer"

_cells, 2022, doi:10.3390/cells11071184_

Round 1

Reviewer 1 Report

Aim of this paper was to profile the protein content of ectosomes released by human thyroid follicular epithelial Nthy-ori 3-1 and anaplastic thyroid carcinoma 8305C cells. The authors identified 568 proteins uniquely expressed by 8305C-derived ectosomes and found that they were able to increase proliferation and motility rate of the recipient cells. The identified ectosome-associated proteins in thyroid cancer cell lines may represent candidate diagnostic and prognostic markers. The study is original but the following critical points should be addressed:

In Fig 6, I suppose the gray dots indicate proteins not involved neither in immune responses or not localized in exosomes. Specify in the panel inside the figure.

It is surprising that the basal wound closure rate (control) is so minimal after 18h in both cell lines. Of note, these rates near 0 were used for normalization, possibly leading to an overestimation of the observed increased closure rate in treated cells. Moreover, the effect of 8305C-derived ectosomes observed on motility rate may be a consequence of increased proliferation and does not necessary indicate an increased migration.

Discussion is too long and should be shortened.

In Tablet 1 angiogenesis category was assigned to more than 15 proteins, as instead stated in Discussion at line 116.

Finally, the authors should consider that Nthy-ori 3-1 cells does not completely reflect normal thyroid cells, maybe primary thyroid cultures would represent a more appropriate model. Moreover, 8305C are derived from the rare anaplastic thyroid carcinoma, and experiments need to be replicated in thyroid cells lines derived from more differentiated thyroid cancer.

Author Response

In Fig 6, I suppose the gray dots indicate proteins not involved neither in immune responses or not localized in exosomes. Specify in the panel inside the figure.

Additional information concerning Figure 6 has been added in the panel inside the figure as suggested.

It is surprising that the basal wound closure rate (control) is so minimal after 18h in both cell lines. Of note, these rates near 0 were used for normalization, possibly leading to an overestimation of the observed increased closure rate in treated cells. Moreover, the effect of 8305C-derived ectosomes observed on motility rate may be a consequence of increased proliferation and does not necessary indicate an increased migration.

The basal wound closure rate after 18 h of incubation was 5% for Nthy-ori 3-1 cells and 4% for 8305C cells. This is actually a very surprising result. None of our previous studies using several different human melanoma cell lines as well as human bladder cancer cell line and human non-malignant ureter epithelium cell line have achieved such minimal wound closure values after 18-24 h of incubation in the control condition. Perhaps this is due to the nature of Nthy-ori 3-1 and 8305C cells, which tend to grow in clusters in culture, unlike other cell lines which grow evenly over the entire surface of the culture dish.

We carefully considered the Reviewer’s comments and agree with the Reviewer's suggestion that with such low values of the wound closure rate used for normalization, this could possibly lead to an overestimation of the observed increased closure rate in ectosome-treated cells. With this in mind, the description of the results obtained in the wound closure test has been modified. Moreover, we agree that it cannot be unequivocally stated that the observed increased wound closure rate in treated cells indicates their increased migration and, in general, increased migratory properties. In control conditions we observed minimal wound closure, suggesting cells of both cell lines are not characterized with high motility. Therefore Reviewer’s suggestion that observed changes in wound closure is more likely related to increased proliferation is justified. In the revised version of the manuscript, we decided to refer to described effects in terms of observation (increased wound closure rate) rather than to the process itself (i.e. increased cell migration). We briefly commented on the matter in the appropriate paragraph in the Results section related to Wound closure assay as cited below:

“Due to the fact that low values of the wound closure rate were used for normalization, the fold values presented in Fig. 10 should be treated as estimates due to the possibility of their slight overestimation, although they show a general tendency.  However, it is also possible that the observed effect of 8305C-derived ectosomes on the mobility rate of recipient cells might have been a consequence of their increased proliferation and not necessarily indicative of their increased migration. Eventually, it could be the result of the collective action of both mentioned factors.”

Moreover, the order of the description of functional tests has also been changed. First, the manuscript describes the results obtained in Alamar Blue Cell Viability Assay, and then in Wound closure assay. Therefore, the numbers of Figures 9 and 10, which showed the results of Wound closure assay and Cell Viability Assay, respectively, have also changed.

Discussion is too long and should be shortened.

We agree with the Reviewer’s point of view that shorter and more concise discussions would make it easier for the reader to focus on the most important points/results of the presented research. Nevertheless, the multiple proteins were identified by us in Nthy-ori 3-1- and 8305C-derived ectosome cargos and these proteins are cited in literature as cancer-related. In our opinion, reference should be made to them in the manuscript. Such thorough description of their role in cancer cell migration, invasion, angiogenesis, immune response etc. may provide an incentive for further studies that would focus on particular molecules from ectosomal cargo. Also, in our Discussion section several potential ectosomal biomarkers of thyroid cancer also mentioned that should be further evaluated regarding their clinical use. Due to the above premises, we decided not to shorten the Discussion.

In Tablet 1 angiogenesis category was assigned to more than 15 proteins, as instead stated in Discussion at line 116.

Thank you very much for this comment and we apologize for the mistake. The number of proteins has been corrected in Discussion at line 116.  The value in both Table 1 and Discussion is 23.

Finally, the authors should consider that Nthy-ori 3-1 cells does not completely reflect normal thyroid cells, maybe primary thyroid cultures would represent a more appropriate model. Moreover, 8305C are derived from the rare anaplastic thyroid carcinoma, and experiments need to be replicated in thyroid cells lines derived from more differentiated thyroid cancer.

We are thankful for the Reviewer’s comment and suggestion regarding the choice of cell lines that were used in the study. Our study was the first to analyzed protein content of thyroid-derived ectosomes (from Nthy-ori 3-1 and 8305C cells), therefore we agree that it should be followed up by studies with wider panel of thyroid cancer cell lines, for instance, representing different disease stages or, as suggested by the reviewer, more differentiated cancer. Primary cultures established from patients’ histologically non-tumorous/tumorous tissues should also be used in future studies to provide more clinically-relevant reference. Since we agree that further studies using other models of TC are needed, we briefly addressed the limitations of our current study in the Conclusions as cited below:

We acknowledge the limitations of our current study. Although our study was the first to analyzed protein content of thyroid-derives ectosomes, two established cell lines that were not derived from the same donor were used in our study. Furthermore, though Nthy-ori 3-1 cells are commonly described as normal human primary thyroid follicular epithelial cells and are regulary used for the study of control and growth in human thyroid, they does not completely reflect normal thyroid cells because they were transfected with a plasmid containing an origin-defective SV40 genome (SV-ori) to immortalize them, however with the retention of several different functions [56]. Thus evaluating changes in ectosome cargo in relation to thyroid cancer with the use of with wider panel of thyroid cancer cell lines, for instance, representing different disease stages or more differentiated cancer, could provide a better understanding of the role of EVs in thyroid cancer progression. Additionally, primary cultures established from patients ’histologically non-tumorous / tumorous tis-sues could also be used in future studies to provide more clinically-relevant reference.”

Reviewer 2 Report

In the manuscript, the authors carried out comparative proteomic profiling of ectosomes derived from thyroid carcinoma and normal thyroid cells by utilizing a shotgun nanoLC-MS/MS approach.  They identified more proteins in the EVs from 8305C cells as compared with those from Nthy-ori-3-1 cells, which might promoter cancer development and progression.  The finding sounds interesting.  There are, however, several concerns as follows.

#1. The authors used only a single cell line for either anaplastic thyroid cancer (8305C cells) or immortalized normal follicular epithelial cells (Nthy-ori-3-1 cells) in the study.  If the authors do not use other cell lines, they cannot conclude that the differentially expressed proteins in the EVs are related to carcer development and progression.  The uniquely expressed proteins such as GDF15 and PRDX5 in the EVs from 8305C cells might be just due to the difference of the cell line. 

In addition, genetic alterations of the cell line should be described in the manuscript. 

#2. The condition of the cell culture should be described in detail.  Cell density, for example, might affect the secretion of the EVs. 

#3. As to the Figure 1., the diameter of the ectosomes is smaller in the Nthy-ori-3-1 cell line than those in the 8305C cell line by TEM analysis (Fig1A), while the particle size of the ectosomes is larger in the Nthy-ori-3-1 cell line than those in the 8305C cell line by NTA analysis (Fig1B).  This seems to be inconsistent.  What is the difference and the reason for the data?

#4. The effects of EVs on cell motility in the wound healing assay seem to be artificial (Figure 9), considering that protein contents are distinct between EVs from 8305C and EVs from Nthy-ori-3-1 (Fig2A).  The authors should, address the mechanisms of the increased motility by the EVs.  They should at least examine the intracellular signaling pathways related to the cell motility.

Author Response

#1. The authors used only a single cell line for either anaplastic thyroid cancer (8305C cells) or immortalized normal follicular epithelial cells (Nthy-ori-3-1 cells) in the study. If the authors do not use other cell lines, they cannot conclude that the differentially expressed proteins in the EVs are related to carcer development and progression. The uniquely expressed proteins such as GDF15 and PRDX5 in the EVs from 8305C cells might be just due to the difference of the cell line.

We are very thankful for the Reviewer’s comment. We fully agree that our pioneering research should be continued using a wider panel of thyroid cancer cell lines, for example representing different stages of the disease or, as the reviewer suggests, more differentiated cancer. Primary cultures established from patients’ histologically non-tumorous/tumorous tissues should also be used in the future studies to provide more clinically-relevant reference. As we agree that further research is needed using other TC models, we briefly address the limitations of our current research in the Conclusions we quote below.

We also agree that the presence of unique proteins in the tumor-derived 8305C ectosomes, with their simultaneous absence in the ectosomes released by Nthy-ori 3-1 cells, might not be the result of malignant  transformation itself, but reflect the specificity of each cell line, as these lines have not been transferred from the same donor. This issue was also addressed in the Conclusions. Nevertheless, the mechanism of protein sorting into ectosomes is strictly regulated and reflects the current state of the parent cell. By this we mean that both cell lines (normal and cancer) can constitutively express a given protein, but only in one of them that protein is selectively sorted into ectosomes. In the case of neoplastic cells, selective segregation of proteins, but also other cargo components, is aimed at inducing changes in the microenvironment through them that will promote tumor progression. We have shown in the Discussion that the role of many of these proteins in cancer development and progression is well characterized, so that their presence in tumor-derived ectosomes is likely to be intentional.

We acknowledge the limitations of our current study. Although our study was the first to analyzed protein content of thyroid-derives ectosomes, two established cell lines that were not derived from the same donor were used in our study. Furthermore, though Nthy-ori 3-1 cells are commonly described as normal human primary thyroid follicular epithelial cells and are regularly used for the study of control and growth in human thyroid, they does not completely reflect normal thyroid cells because they were transfected with a plasmid containing an origin-defective SV40 genome (SV-ori) to immortalize them, however with the retention of several different functions [56]. Thus evaluating changes in ectosome cargo in relation to thyroid cancer with the use of with wider panel of thyroid cancer cell lines, for instance, representing different disease stages or more differentiated cancer, could provide a better understanding of the role of EVs in thyroid cancer progression. Additionally, primary cultures established from patients ’histologically non-tumorous / tumorous tis-sues could also be used in future studies to provide more clinically-relevant reference.”

In addition, genetic alterations of the cell line should be described in the manuscript.

The required information was provided in Methods. Nthy-ori 3-1 cells were immortalized by transfection with 1 μg of the plasmid SV40ori (large T antigen)– comprising of 5.3 kB SV40 genome with a six base pair dele-tion that eliminated the BglI site at the origin of replication cloned in pMK16 [21]. T antigen interacts with two protein suppresors of cell division (pRB1 and p53) and inhibits their activity, thus enabling unlimited number of divisions of transfected Nthy-ori 3-1 cells. Information on genetic alterations of the cell lines are provided in [22, 23].

#2. The condition of the cell culture should be described in detail. Cell density, for example, might affect the secretion of the EVs.

We agree with the Reviewer that the way of carrying out of the cell culture affects EV secretion. In the manuscript the basic conditions in which cells were culture before ectosome isolation (cell culture media, supplementation, temperature etc.) were already described in Methods. Regarding the cell density, we provided an additional information on the cell culture confluence (80%) that must have been reached before ectosome isolation and the number of Petri Dishes that was used to collect conditioned media. To obtain the precise cell density (e.g. in cell/ml), the cells would have to be trypsinized and counted post factum - after conditioned media is collected for ectosome isolation, when any adjustments to the culture are no longer possible. In our opinion, information about percentage of confluence and the number of required dishes is more useful. Choosing the particular confluence value as a starting point to collect media for ectosome isolation and following it through all experiments allows us to obtain similar and sufficient amount of ectosomes.

#3. As to the Figure 1., the diameter of the ectosomes is smaller in the Nthy-ori-3-1 cell line than those in the 8305C cell line by TEM analysis (Fig1A), while the particle size of the ectosomes is larger in the Nthy-ori-3-1 cell line than those in the 8305C cell line by NTA analysis (Fig1B).  This seems to be inconsistent.  What is the difference and the reason for the data?

Indeed, the sole mean values of ectosomes diameter from NTA experiments for Nthy-ori 3-1- and 8305C-derived are the reverse of results from TEM analysis. However, when considering standard deviations for those means, there are no significant differences, meaning that ectosomes from both cell lines did not differ in size (i.e. diameter) according to both TEM and NTA measurements.

#4. The effects of EVs on cell motility in the wound healing assay seem to be artificial (Figure 9), considering that protein contents are distinct between EVs from 8305C and EVs from Nthy-ori-3-1 (Fig2A).  The authors should, address the mechanisms of the increased motility by the EVs.  They should at least examine the intracellular signaling pathways related to the cell motility.

We considered the Reviewer’s comments and agree that it cannot be unequivocally stated that the observed increased wound closure rate in treated cells indicates their increased migration and, in general, increased migratory properties. In control conditions we observed minimal wound closure, suggesting cells of both cell lines are not characterized with high motility. Therefore Reviewer’s suggestion that observed changes in wound closure is more likely related to increased proliferation is justified. In the revised version of the manuscript, we decided to refer to described effects in terms of observation (increased wound closure rate) rather than to the process itself (i.e. increased cell migration). We briefly commented on the matter in the appropriate paragraph in the Results section related to wound closure assay as cited below:

“it is important to notice that the basal wound closure rate (in untreated cells) is minimal (approx. 4-5%) after 18h in both cell lines. That might have led to a slight overestimation of the observed effects in treated cells. Also, the observed increased wound closure rates effect might have been a consequence of increased proliferation and do not necessary indicate an increased migration. Eventually, it was the result of the collective action of both mentioned factors.”

Regarding distinct contents of EVs from 8305C and from Nthy-ori-3-1 cells, functional tests (wound healing and Alamar Blue assays) were already performed using both cell lines as recipient cells and both as ectosome donors. It allowed us to include all potential ectosome-recipient cell interaction variants and observed all possible effects exerted by both normal cell- and tumor cell-derived ectosomes with respect to their different cargos.

Regarding intracellular signaling pathways related to the cell motility, in the Discussion section 4.1. we describe multiple proteins involved in signaling pathways related to cancer cell motility, including integrin signaling, GDF15-STAT3 signaling axis, Src kinase and several more. All those proteins were identified in ectosomes by our study as well as in other thyroid cancer-oriented research that we cited in that paragraph. Detailed analysis of the chosen, single signaling pathway of interest should be considered for further studies.

Round 2

Reviewer 1 Report

Published data reported a higher basal wound closure  after 18/24h both in Nthy-ori and 8305C celll lines (Bravo-Miana et al. 2020, Coperchini et al. 2019). After excluding mycoplasma contamination or other procedural problems (e.g. coating removal with the tip) I suggest the authors  to avoid normalization of the wound closure rate to the basal condition. 

Author Response

Response to Reviewer 1

Published data reported a higher basal wound closure  after 18/24h both in Nthy-ori and 8305C celll lines (Bravo-Miana et al. 2020, Coperchini et al. 2019). After excluding mycoplasma contamination or other procedural problems (e.g. coating removal with the tip) I suggest the authors  to avoid normalization of the wound closure rate to the basal condition. 

As suggested by the Reviewer we corrected Figure 10. The results are now presented in percentage of wound closure instead of being normalized to untreated control. Regarding the mentioned literature, both papers analyzed the results of wound healing assay in different manner i.e. total wound area. Since we presented our results as change in wound width, they cannot be directly compared.

Also we would like to reassure that the wound was created with the highest caution and no cell coating was removed in the process. Also all the cells were routinely tested for mycoplasma.

Reviewer 2 Report

My concerns are appropriately addressed by the comments from the authors. 

Author Response

Response to Reviewer 2

My concerns are appropriately addressed by the comments from the authors. 

We would like to thank the Reviewer for accepting our Revision and for all previous comments that helped to improve our manuscript.